# Echo state network model for analyzing solar-wind effects on the $AU$ and $AL$ indices

Shin'ya Nakano[1,2,3] and Ryuho Kataoka[4,3]

[1]The Institute of Statistical Mathematics, Tachikawa, 190–8562, Japan
[2]Center for Data Assimilation Research and Applications, Joint Support Center for Data Science Research, Tachikawa, Japan.
[3]School of Multidisciplinary Science, SOKENDAI, Hayama, 240–0115, Japan.
[4]National Institute of Polar Research, Tachikawa, Japan.

**Correspondence:** Shin'ya Nakano (shiny@ism.ac.jp)

**Abstract.** The properties of the auroral electrojets are examined on the basis of a trained machine learning model. The relationships between solar-wind parameters and the $AU$ and $AL$ indices are modeled with an echo state network (ESN), a kind of recurrent neural network. We can consider this trained ESN model to represent nonlinear effects of the solar-wind inputs on the auroral electrojets. To identify the properties of auroral electrojets, we obtain various synthetic $AU$ and $AL$ data by using various artificial inputs with the trained ESN. The analyses of various synthetic data show that the $AU$ and $AL$ indices are mainly controlled by the solar-wind speed in addition to $B_z$ of the interplanetary magnetic field (IMF) as suggested by the literature. The results also indicate that the solar-wind density effect is emphasized when solar-wind speed is high and when IMF $B_z$ is near zero. This suggests some nonlinear effects of the solar-wind density.

## 1 Introduction

Auroral electrojets are azimuthal electric currents localized in the auroral region. A westward auroral electrojet is mostly observed in pre-midnight to early morning local time, and an eastward electrojet is mostly observed in evening time (Allen and Kroehl, 1975). The $AU$ and $AL$ indices (Davis and Sugiura, 1966; World Data Center for Geomagnetism, Kyoto et al., 2015) represent the strengths of eastward and westward electrojets, respectively, and are widely used for monitoring geomagnetic activity in the auroral region. It is widely accepted that the behavior of the auroral electrojet is mainly controlled by the solar wind input into the magnetosphere. In particular, many studies suggest that the southward component of the interplanetary magnetic field (IMF) and the solar-wind speed have essential effects on auroral activity as measured by $AU$ and $AL$ indices (e.g., Akasofu, 1981; Murayama, 1982). The solar-wind density is also likely to contribute to the auroral electrojet intensity (e.g., Newell et al., 2007; Ebihara et al., 2019). However, multiple physical processes can contribute to the development of the auroral indices, and some of the processes are nonlinear to the solar-wind input (e.g., Clauer and Kamide, 1985; Kamide and Kokubun, 1996). Hence it is not a simple task to model the temporal evolution of the $AU$ and $AL$ indices.

To describe the complicated processes of the indices, Luo et al. (2013) constructed a parametric model with many parameters. Machine learning approaches are also used in many studies to describe the nonlinear evolution of the auroral electrojets. For example, Chen and Sharma (2006) employed the weighted nearest neighbors method for predicting the $AL$ index during

storm times. In particular, artificial neural networks are frequently used for modeling the $AU$, $AL$, and $AE$ indices. It has been demonstrated that the $AU$, $AL$, and $AE$ indices can be predicted well with feed-forward neural networks using time histories of solar-wind parameters as inputs (e.g., Gleisner and Lundstedy, 1997; Takalo and Timonen, 1997; Pallocchia et al., 2008; Bala and Reiff, 2012). Recurrent types of neural networks are also useful for representing dynamical behaviors of the magnetosphere (Gleisner and Lundstedy, 2001). Amariutei and Ganushkina (2012) predicted the $AL$ index using a model which combines the autoregressive moving average with the exogenous inputs (ARMAX) model and a neural network.

While machine learning techniques tend to be used for predictions with high accuracy, the learned relationships between solar-wind inputs and auroral electrojets are of interest from the scientific perspective as well. Since most machine learning models such as neural networks are nonlinear model, trained machine learning models can describe the nonlinear behaviors of the magnetospheric system. It is thus meaningful to analyze the input–output relationships of the trained machine learning models. Recently, Blunier et al. (2021) have identified solar-wind parameters which affect the value of geomagnetic indices by putting perturbed inputs into a trained neural network. This study takes a somewhat similar approach. We employ an echo state network (ESN) model (Jaeger, 2001; Jaeger and Haas, 2004) to describe the relationship between various solar-wind parameters and the auroral electrojet indices $AU$ and $AL$. The ESN is a kind of recurrent neural network, which can be used for describing nonlinear systems (e.g., Chattopadhyay et al., 2020). We then examine the responses of the $AU$ and $AL$ indices to solar-wind inputs by putting various artificial inputs into the trained ESN model and identify the properties of the auroral electrojets.

## 2 Echo state network

We model the temporal evolution of $AU$ and $AL$ with the ESN model because it can be easily implemented to attain a satisfactory performance. The ESN is a recurrent neural network with fixed random connections and weights between hidden state variables. Only the weights for the output layer are trained so that the target temporal pattern is well reproduced. We combine the state variables at the time $t_k$ into a vector $\boldsymbol{x}_k$, where the $i$-th element of $\boldsymbol{x}_k$ is denoted as $x_{k,i}$. The number of state variables $m$ is set at 1200 in this study. At the time step $k$, we update $x_{k,i}$ as follows:

$$x_{k,i} = (1 - \xi)x_{k-1,i} + \xi \tanh\left(\boldsymbol{w}_i^T \boldsymbol{x}_{k-1} + \boldsymbol{u}_i^T \boldsymbol{z}_k + \eta_i\right) \tag{1}$$

where $\boldsymbol{z}_k$ is a vector consisting of the input variables. The parameter $\xi$ is the leaking rate (Jaeger et al., 2007; Lukoševičius, 2012) and its value is fixed at 0.5 in this paper. The weights $\boldsymbol{w}_i$ and $\boldsymbol{u}_i$ determine the connection with the other state variables and input variables. The weights $\boldsymbol{w}_i$ and the parameter $\eta_i$ are given in advance and are fixed.

It is desirable that the weights are given so as to attain the so-called 'echo state property'. The echo state property guarantees that the ESN forgets distant past inputs. Defining the weight matrix $W$ as

$$W = (\boldsymbol{w}_1 \; \boldsymbol{w}_2 \; \cdots \boldsymbol{w}_m), \tag{2}$$

a sufficient condition for the echo state property is that the maximum singular value of $W$ is less than 1. If a certain matrix $W'$ is given and its maximum singular value $\lambda'$ is computed, we can obtain the weight matrix $W$ which satisfies this sufficient

condition as follows:

$$W = \frac{\alpha}{\lambda'} W'. \tag{3}$$

We thus first determine $W'$ randomly and obtain the weight $W$ according to Eq. (3) with the parameter $\alpha$ set to 0.99. In this study, we set 90% of the elements of $W'$ to be zero. Each of the remaining non-zero elements comprising 10% of $W'$ is obtained randomly from a Laplace distribution for which the probability density function $p(x)$ is written as

$$p(x) = \frac{1}{2} \exp\left(-|x|\right). \tag{4}$$

Similarly to $W'$, 90% of the elements of $\boldsymbol{u}_i$ are set to be zero and the other non-zero elements are given by the same Laplace distribution. The parameter $\eta_i$ in Eq. (1) is obtained randomly from a normal distribution with mean 0 and standard deviation 0.3.

The output for the time $t_k$, $\boldsymbol{y}_k$, is obtained from $\boldsymbol{x}_k$ as follows:

$$\boldsymbol{y}_k = \boldsymbol{\beta}^T \boldsymbol{x}_k. \tag{5}$$

The weight $\boldsymbol{\beta}$ in Eq. (5) is determined so that the objective function

$$J = \sum_{k=1}^{K} \|\boldsymbol{d}_k - \boldsymbol{y}_k\|^2 \tag{6}$$

is minimized, where $\boldsymbol{d}_k$ is an observation vector consisting of the observed data. The present study aims to model the temporal pattern of the $AU$ and $AL$ indices. Accordingly, the output vector $\boldsymbol{y}_k$ consists of two elements as follows

$$\boldsymbol{y}_k = \begin{pmatrix} y_{AU,k} \\ y_{AL,k} \end{pmatrix}, \tag{7}$$

where $y_{AU,k}$ and $y_{AL,k}$ are the predicted $AU$ and $AL$ values at $t_k$, respectively. In this study, 5-minute values (averages for 5 minutes) of $AU$ and $AL$ are used. We give the input vector $\boldsymbol{z}_k$ as follows:

$$\boldsymbol{z}_k = \begin{pmatrix} B_{x,k}/S_{B_x} \\ B_{y,k}/S_{B_y} \\ B_{z,k}/S_{B_z} \\ (V_{sw,k} - b_V)/S_V \\ (N_{sw,k} - b_N)/S_N \\ (T_{sw,k} - b_T)/S_T \\ \cos\left(2\pi H_k/24\right) \\ \sin\left(2\pi H_k/24\right) \\ \cos\left(2\pi D_k/364.24\right) \\ \sin\left(2\pi D_k/364.24\right) \\ y_{AU,k-1}/S_{AU} \\ y_{AL,k-1}/S_{AL} \end{pmatrix} \tag{8}$$

where $B_{x,k}$, $B_{z,k}$ and $B_{y,k}$ denote the $x$, $y$, and $z$ component of the interplanetary magnetic field in the geocentric solar magnetic (GSM) coordinates at time $t_k$, $V_{sw,k}$ is the $-x$ component of the solar wind velocity in the GSM coordinates, $N_{sw,k}$ is the solar wind density, $T_{sw,k}$ is the solar wind temperature, $H_k$ is universal time (UT) in hour, and $D_k$ is the day from the end of 2000 ($D_k = 1$ on January 1, 2001). $S_{B_x}$, $S_{B_y}$, $S_{B_z}$, $S_V$, $S_N$, $S_T$, $S_{AU}$, and $S_{AL}$ are rescaling factors to adjust the value of each element of $\boldsymbol{z}_k$ to a similar range, and $b_V$, $b_N$, and $b_T$ are also for adjusting the range of each element of $\boldsymbol{z}_k$. We set $S_{B_x} = S_{B_y} = S_{B_z} = 10\,(\mathrm{nT})$, $S_V = 500\,(\mathrm{km/s})$, $S_N = 20\,(/\mathrm{cc})$, $S_T = 10^6\,(\mathrm{K})$, $S_{AU} = S_{AL} = 1000\,(\mathrm{nT})$, $b_V = 400\,(\mathrm{km/s})$, $b_N = 1\,(/\mathrm{cc})$, and $b_T = 2 \times 10^5\,(\mathrm{K})$. The variables $H_k$ and $D_k$ are included for considering UT dependence and seasonal dependence (e.g., Cliver et al., 2000). The feedback of the predicted $AU$ and $AL$ indices which can be obtained using Eq. (5) is also included in the input vector $\boldsymbol{z}_k$. The solar wind variables $B_{x,k}$, $B_{y,k}$, $B_{z,k}$, $V_{sw,k}$, $N_{sw,k}$, and $T_{sw,k}$ are taken from the OMNI 5-minute data.

If $\boldsymbol{z}_k$ does not contain the feedback of $y_{AU,k-1}$ and $y_{AL,k-1}$, the weight $\boldsymbol{\beta}$ can be determined through simple linear regression because $\boldsymbol{x}_k$ at each time step would not depend on $\boldsymbol{\beta}$ in Eq. (5). However, since the feedback of $y_{AU,k-1}$ and $y_{AL,k-1}$ are contained, the optimal $\boldsymbol{\beta}$ cannot be obtained by linear regression. We thus obtained $\boldsymbol{\beta}$ using the ensemble-based optimization method (Nakano, 2021).

## 3 Performance of ESN

We trained the ESN using data obtained over a period of ten years from 2005 to 2014. We used 5-minute values of the OMNI solar wind data and the $AU$ and $AL$ indices provided by Kyoto University. Since each of the state variables of the ESN is obtained by a nonlinear conversion of the previous state variables according to Eq. (1), the ESN memorizes the history of the input data. When predicting the $AU$ and $AL$ indices, the ESN requires the solar-wind data for the preceding several time steps. Hence we start the comparison after spin-up of the ESN for 72 steps, which corresponds to 6 hours for the 5-minute values, from the initial time of the analysis. It should also be noted that solar wind data are sometimes incomplete. If more than half of the data were missing for 1 hour, we stopped the prediction and spun up the ESN again for the subsequent 72 steps.

We then reproduced the $AU$ and $AL$ indices for the period from 1998 to 2004 and compared the outputs with the observed values. In Figure 1, the top panel shows the $AU$ and $AL$ reproduced by our ESN model in October 1999 with red lines and the observed $AU$ and $AL$ indices with gray lines for the same period. The second panel shows the three components of the IMF. The green, blue, and red lines indicate the $x$, $y$, and $z$ components in (GSM) coordinates, respectively. The third panel shows the solar wind speed and the fourth panel shows the solar wind density. The bottom panel shows the *SYM-H* index (Iyemori, 1990; Iyemori and Rao, 1996) for the corresponding time period. High auroral activity was maintained for the period from 10 October to 17 October when high speed solar wind streams coincided with a continual southward IMF, as suggested by the literature (e.g., Tsurutani et al., 1990, 1995). The auroral activity was also enhanced during a magnetic storm from 21 October. The model outputs mostly reproduced the observed $AU$ and $AL$ values well for these events.

Table 1 shows the root-mean-square errors (RMSE) of the ESN prediction for each year of the period from 1998 to 2004. The Pearson correlation coefficients between the ESN prediction and the observation are also indicated in this table. The RMSEs

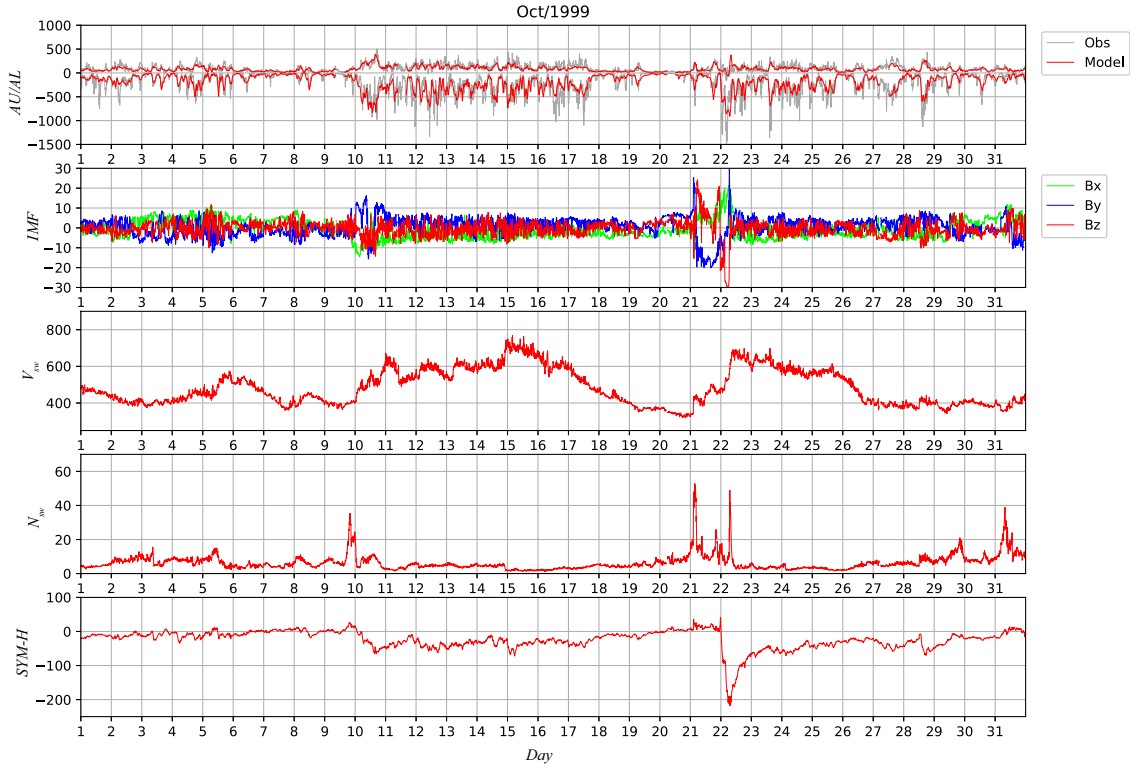

**Figure 1.** The top panel shows the $AU$ and $AL$ values for October 1999 reproduced with the ESN model (red) and the observed $AU$ and $AL$ indices (gray). The second panel shows the IMF $B_x$ (green), $B_y$ (blue), and $B_z$ (red) in GSM coordinates. The third panel shows the solar wind speed, the fourth panel shows the solar wind density, and the bottom panel shows the *SYM-H* index.

were less than 100 nT for the $AL$ index and about 50 nT for the $AU$ index except for 2003. The RMSEs of $AU$ and $AL$ were larger in 2003 than in other years likely because of high auroral activity during that year. Figure 2 shows the means of the $|AU|$ and $|AL|$ values for each month from 1998 to 2004. The mean of $|AL|$ exceeded 200 nT in most of the months in 2003, which indicates high activity of the westward auroral electrojet. The mean of $|AU|$ also tended to be larger in 2003 than in the other years. The correlation coefficients were around 0.8 for both $AU$ and $AL$ over the period shown in this table. In the model of Luo et al. (2013), which predicted the 10-minute values of the $AE$ indices from solar wind parameters, the RMSEs were 83.8, 125.5, and 102.0 nT in 2002, 2003, and 2004, respectively, for the $AL$ index and 44.5, 58.7, and 47.7 nT in 2002, 2003, and 2004 for the $AU$ index. Our ESN model thus achieves an accuracy comparable to the model of Luo et al.. While Luo et al. used 10-minute values, this study uses 5-minute values in the prediction. Considering that data with a higher time resolution tend to contain larger noise, we believe that the ESN meets a satisfactory accuracy in comparison with other existing models.

**Table 1.** The root-mean-square errors of the ESN prediction (in nT) and the Pearson correlation coefficients between the ESN prediction and the observation for the $AL$ and $AU$ indices.

| Year | RMSE ($AL$) | Corr. coef. ($AL$) | RMSE ($AU$) | Corr. coef. ($AU$) |
|------|-------------|--------------------|-------------|--------------------|
| 1998 | 91.21       | 0.85               | 44.67       | 0.84               |
| 1999 | 88.00       | 0.84               | 47.06       | 0.82               |
| 2000 | 99.10       | 0.82               | 58.20       | 0.82               |
| 2001 | 96.75       | 0.81               | 53.36       | 0.81               |
| 2002 | 89.90       | 0.83               | 50.52       | 0.82               |
| 2003 | 118.62      | 0.82               | 63.50       | 0.77               |
| 2004 | 99.84       | 0.84               | 47.72       | 0.78               |

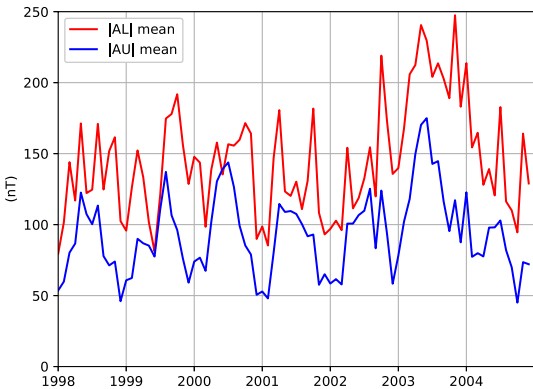

**Figure 2.** The means of $|AU|$ and $|AL|$ for each month from 1998 to 2004.

## 110    4    Responses to synthetic solar wind

Machine learning models including the ESN model can be regarded as nonlinear regression models for summarizing the relationship between an input and an output. As the ESN model is a 'black-box' model, we cannot directly extract the input-output relationships in a functional form. However, we can experimentally examine the responses of the $AU$ and $AL$ indices to various solar-wind inputs by using the trained ESN model. If we put artificial inputs into the trained ESN model, we obtain

synthetic $AU$ and $AL$ indices as outputs of the model under the given inputs. We can then identify properties of the auroral electrojets by analyzing the synthetic indices obtained from various artificial inputs.

We obtained synthetic $AU$ and $AL$ indices by the ESN with an artificial input where the value of one of the solar-wind parameters was fixed. For example, we turned off the variation of IMF $B_x$ by fixing it at a constant 0 nT and derived synthetic $AU$ and $AL$ indices where the $B_x$ effect was excluded. We then compared the synthetic indices with the observed indices for

each year to evaluate the impact of IMF $B_x$. Similarly, we obtained synthetic indices which exclude each of the effects of IMF $B_y$, solar-wind speed, solar-wind density, and solar-wind temperature, and evaluated the impact of each parameter for each year. The fixed values of IMF $B_y$, solar-wind speed, solar-wind density, and solar-wind temperature were $0\,\mathrm{nT}$, $400\,\mathrm{km/s}$, $1/\mathrm{cc}$, and $2 \times 10^5\,\mathrm{K}$, respectively. We did not consider the case where the IMF $B_z$ effect was turned off because the RMSE becomes very large without an accurate IMF $B_z$ input, as obviously expected from the results of many previous studies (e.g.,

Arnoldy, 1971; Akasofu, 1981; Murayama, 1982; Newell et al., 2007).

Figures 3 and 4 show the RMSE and mean deviation values in each year for the various synthetic $AL$ indices where the effect of one of the solar-wind parameters was excluded. In each figure, the red lines show the RMSEs for the output of ESN using all the solar-wind parameters described in Eq. (8). The green and blue lines show the RMSEs when the effects of IMF $B_x$ and $B_y$ were excluded, respectively. The orange, light blue, and gray lines show the respective RMSEs when the effects of solar-

wind speed, density, and temperature were excluded. To evaluate the uncertainty, we prepared 10 data sets, each of which was obtained by leaving out the data for one of the ten years from 2005 to 2014 and calculated the weights $\boldsymbol{\beta}$ in Eq. (5) using each of the 10 data sets. We then obtained the synthetic $AU$ and $AL$ indices using the ESN with each of these different 10 weight values. The solid lines in Figure 3 and 4 show the mean values for the 10 synthetic $AL$ indices. The dashed lines indicate the maxima and minima among the 10 outputs. Among the six solar-wind parameters, the effect of solar-wind speed is prominent

especially in 2003, when some severe magnetic storms were observed, presumably because it contributes to the efficiency of the coupling between the solar wind and the Earth's magnetosphere (e.g., Akasofu, 1981; Murayama, 1982; Newell et al., 2007). The mean deviation shown in Figure 4 indicates the bias of the ESN output, and the positive bias means that the ESN output tends to be larger than the observed $AL$ value, which corresponds to an underestimation of $|AL|$. The large positive bias for the case without solar-wind speed variation in Figure 4 thus suggests that a low solar-wind speed results in a small $|AL|$.

Conversely, a high solar-wind speed activates variations of $AL$. We can also observe a relatively small effect of IMF $B_y$, which would also contribute to the coupling between the solar wind and the magnetosphere. In addition, the effect of the solar-wind density can be seen for all of the years from 1998 to 2004. Figure 5 extracts the RMSEs for the case without the IMF $B_y$ effect and the case without the solar-wind density effect from Figure 3 and compares them with the case with all the solar-wind parameters in an expanded scale. This demonstrates that the effects of IMF $B_y$ and the solar-wind density on the RMSEs are

mostly larger than the scale of the uncertainty. The large mean deviation suggests that the solar-wind density enhancement intensifies the westward electrojet as implied by some earlier studies (Newell et al., 2008; McPherron et al., 2015).

Figures 6 and 7 show the RMSE and the mean deviation values for the various synthetic $AU$ indices. Each color indicates the result with the same input as the corresponding color in Figure 3. The solar-wind speed effect is again prominent. The large negative bias for the case without solar-wind speed variation in Figure 7 suggests a low solar-wind speed underestimates the

$AU$ value. In contrast with $AL$, $AU$ is likely to be strongly controlled by IMF $B_y$ and the solar-wind density. In particular, the mean deviation is largely negative for the case without density variation, which suggests an important effect of solar-wind density on the $AU$ index, as discussed by Blunier et al. (2021).

The top panel in Figure 8 shows some of the synthetic $AU$ and $AL$ indices from 21 October to 25 October in 1999. The red lines indicate the output where all of the parameters in Eq. (8) were used. The green and blue lines indicate the synthetic values

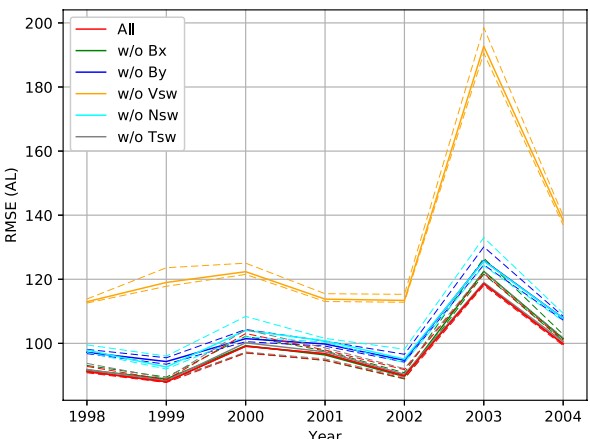

**Figure 3.** RMSE in each year for the various synthetic $AL$ indices where the effect of one of the solar-wind parameters was excluded.

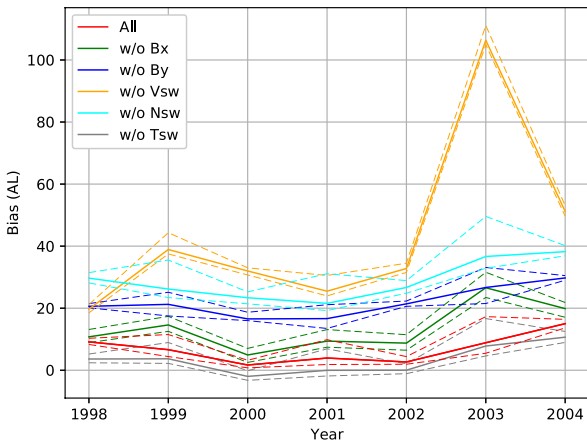

**Figure 4.** Mean deviation in each year for the various synthetic $AL$ indices where the effect of one of the solar-wind parameters was excluded.

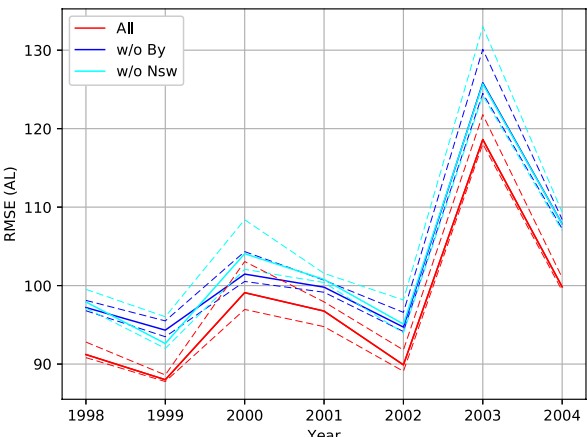

**Figure 5.** RMSE in each year for the various synthetic $AL$ indices where the effect of one of the solar-wind parameters was excluded.

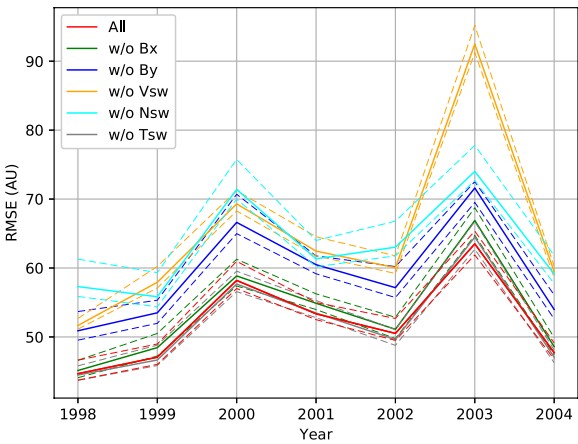

**Figure 6.** RMSE in each year for the various synthetic $AU$ indices where the effect of one of the solar-wind parameters was excluded.

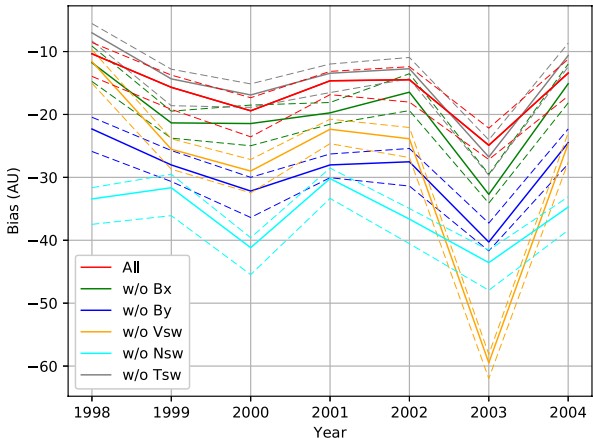

**Figure 7.** Mean deviation in each year for the various synthetic $AU$ indices where the effect of one of the solar-wind parameters was excluded.

where solar-wind speed and density was turned off, respectively. The gray lines show the observed actual $AU$ and $AL$ indices for reference. The other panels in this figure are the same as those in Figure 1. Although the ESN output is much smoother than the observation, especially in some impulsive events which would be related to substorms, the red line reproduces the observed $AU$ and $AL$ indices well. In contrast, when the solar-wind speed was set to be low at $400\,\mathrm{km/s}$, the ESN model clearly underpredicted the strength of $AL$. This suggests that a high-speed solar wind makes an important contribution to enhancing the westward electrojet. When the density effect was turned off, the ESN tended to slightly underpredict $|AL|$ although the density effect was likely to be minor in this event.

Figure 9 shows the result for another event from 26 July to 30 July in 2000. In this event, since the solar-wind speed was maintained at around $400\,\mathrm{km/s}$, which we set as the base level of the solar-wind speed, the green line was similar to the red line. On the other hand, the solar-wind density effect is visible. If the density is fixed at $1/\mathrm{cc}$, the ESN tended to underpredict $|AU|$ and $|AL|$. However, the relationships with the solar-wind density learned by the ESN seemed to not be linear. For example, the difference between the red and blue lines tended to be larger on 29 July than on 28 July while the solar-wind density was more enhanced on 28 July than on 29 July. This might suggest some compound effects of the solar-wind density and other parameters.

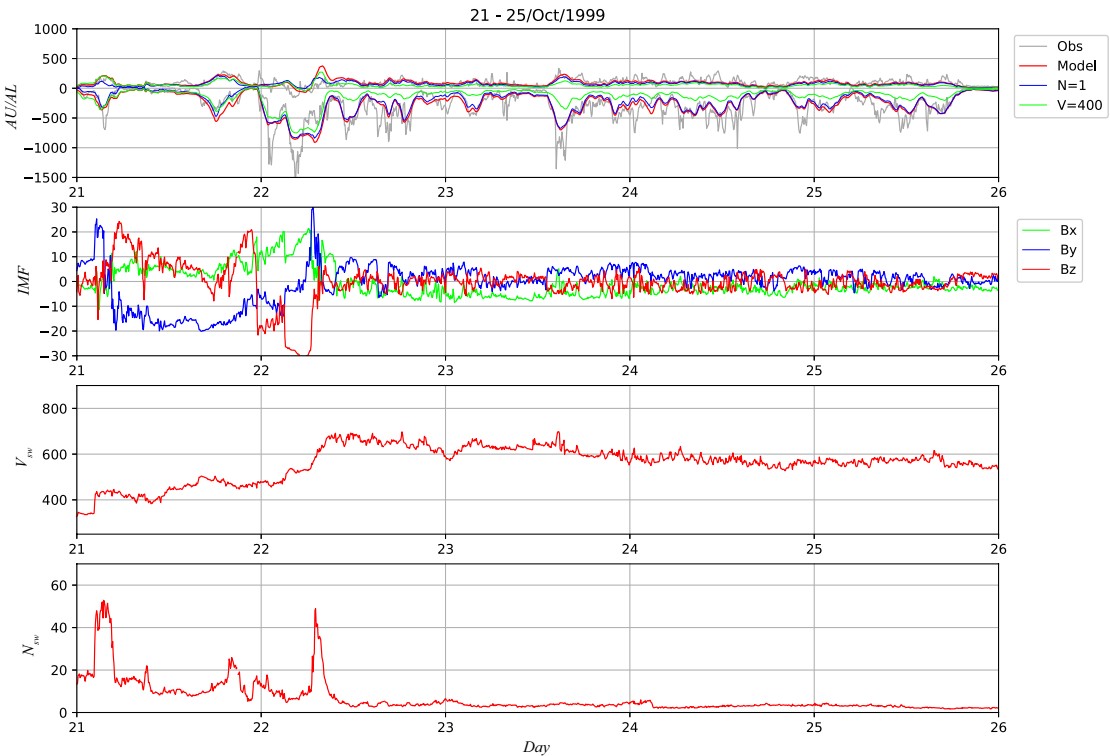

**Figure 8.** Comparison of some ESN outputs during the period from 21 October to 25 October 1999. The top panel shows the ESN output with all the parameters (red), the synthetic indices where the solar-wind speed effect was turned off (green), those where the solar-wind density effect was turned off (blue), and the observed $AU$ and $AL$ indices (gray). The second panel shows the IMF $B_x$ (green), $B_y$ (blue), and $B_z$ (red) in GSM coordinates. The third panel shows the solar wind speed, the fourth panel shows the solar wind density, and the bottom panel shows the *SYM-H* index.

We closely examined the density effects learned by the ESN by computing other synthetic indices $AU(N = 20)$ and $AL(N = 20)$, obtained by fixing the solar-wind density input of the ESN at $20\,/\mathrm{cc}$. We then obtained the differences

$$\Delta AU_{N\mathrm{eff}} = AU(N = 20) - AU(N = 1),$$
$$\Delta AL_{N\mathrm{eff}} = AL(N = 20) - AL(N = 1)$$

where $AU(N = 1)$ and $AL(N = 1)$ are the synthetic $AU$ and $AL$ indices obtained by fixing the solar-wind density at $1\,/\mathrm{cc}$. We then used $\Delta AU_{N\mathrm{eff}}$ and $\Delta AL_{N\mathrm{eff}}$ as proxies of the solar-wind density effect as a function of time. Figure 10 is a 2-dimensional histogram to compare $\Delta AU_{N\mathrm{eff}}$ and $\Delta AL_{N\mathrm{eff}}$ with the solar-wind speed. As the solar-wind speed increases, $\Delta AU_{N\mathrm{eff}}$ increases and $\Delta AL_{N\mathrm{eff}}$ decreases. This suggests that the solar-wind density effect on the auroral electrojets is not independent of the solar-wind speed effect but that the solar-wind density contributes to the auroral electrojet intensity more effectively under high solar-wind speed conditions. The solar-wind density effect is likely to be small when the solar-wind

speed is low. Figure 11 is a 2-dimensional histogram to compare $\Delta AU_{N\text{eff}}$ and $\Delta AL_{N\text{eff}}$ with IMF $B_z$. The solar-wind density effect gets large when IMF $B_z$ is near zero. The density effect is small on average when $|B_z|$ is large. The ESN model therefore suggests that the solar-wind density effect is most important when IMF $B_z$ is small.

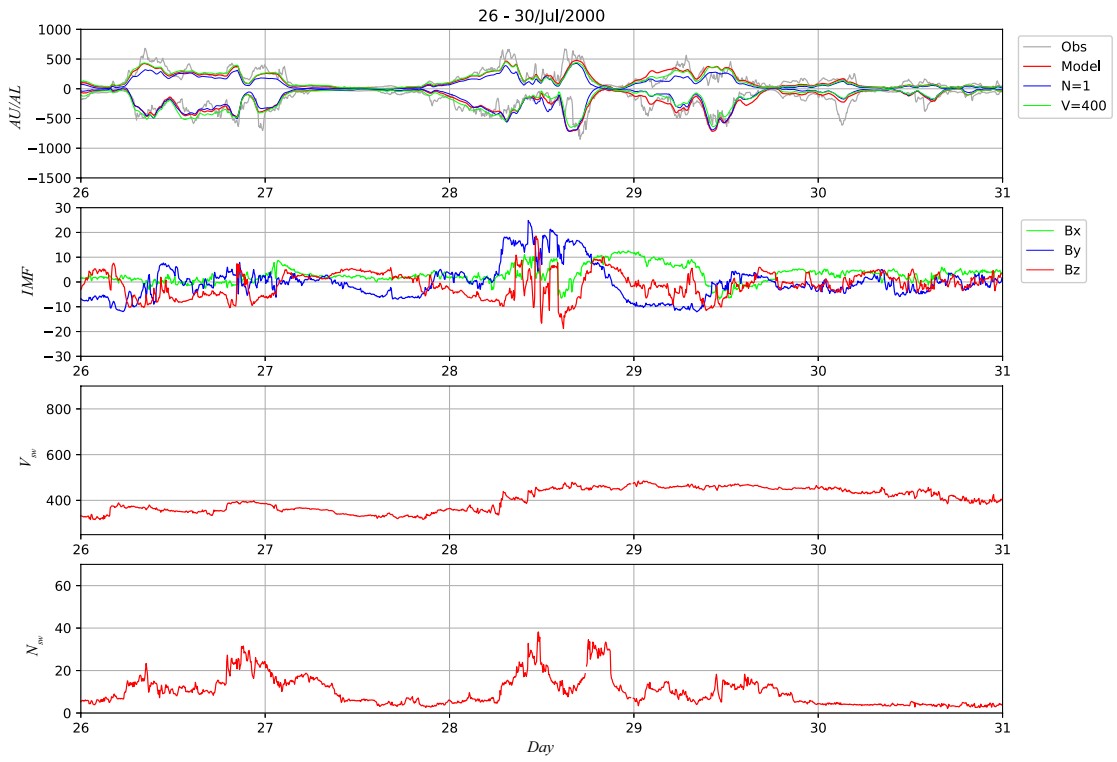

**Figure 9.** Comparison of ESN outputs during the period from 26 July to 30 July 2000 in the same format as Figure 8.

We also conducted an experiment in which the solar wind parameters are fixed at constant values except that one of the parameters are given by rectangular waves with various periods. Figure 12 shows the result of this experiment. IMF $B_x$ and $B_y$ were set at $0$ and the temperature was fixed at $5 \times 10^5\,\text{K}$ through this experiment. In the first six days, IMF $B_z$ was perturbed with a rectangular wave with a period of 20 minutes for the first two days, 2 hours for the second two days, and 6 hours for the third two days, while the solar-wind speed was fixed at $400\,\text{km/s}$ and the density was fixed at $2\,/\text{cc}$. In the next six days,

IMF $B_z$ was perturbed with the same pattern but the solar-wind speed was changed at $800\,\text{km/s}$. After that, IMF $B_z$ was fixed at $-5\,\text{nT}$ and the solar-wind speed was perturbed with a similar rectangular pattern for six days. The solar-wind speed was then fixed at $800\,\text{km/s}$, and the solar-wind density was perturbed with a similar rectangular pattern under the fixed IMF $B_z$ at $1\,\text{nT}$ and $-5\,\text{nT}$. The ESN output shown in the upper panel exhibits daily variations, which are due to the UT dependence considered in Eq. (8). Although the ESN output tends to be smoother than the observed variation as shown in Figures 8 and 9,

the effects of the perturbations with a period of at least 2 hours are observed in the temporal patterns of the auroral electrojets.

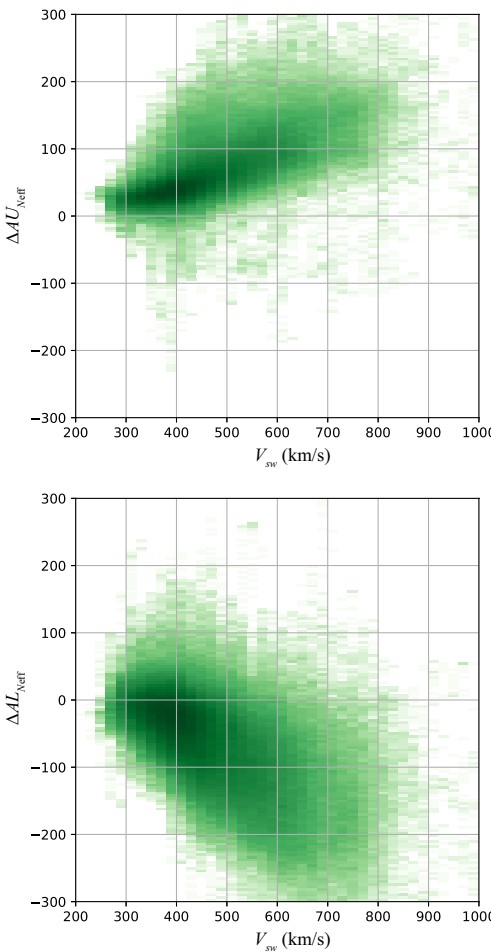

**Figure 10.** 2-dimensional histogram indicating the dependence of the solar-wind density effect on the solar-wind speed.

The response to the solar-wind density variations is clearer when IMF $B_z$ is $1\,\mathrm{nT}$ than when it is $5\,\mathrm{nT}$, which is consistent with the result shown in Figure 11.

## 5 Discussion

It is widely accepted that auroral electrojets are mainly controlled by IMF and the solar-wind speed (e.g., Akasofu, 1981; Murayama, 1982; Newell et al., 2007). In particular, IMF $B_z$ has an essential effect on auroral activity. When IMF is directed southward, DP2 type electrojets (e.g., Kamide and Kokubun, 1996) are enhanced and contribute to both $AU$ and $AL$. The substorm current wedge, which contains a westward electrojet contributing to the $AL$ index, would also be controlled by IMF (e.g., Kepko et al., 2015). As illustrated in Figure 1, the solar-wind speed also has an important effect.

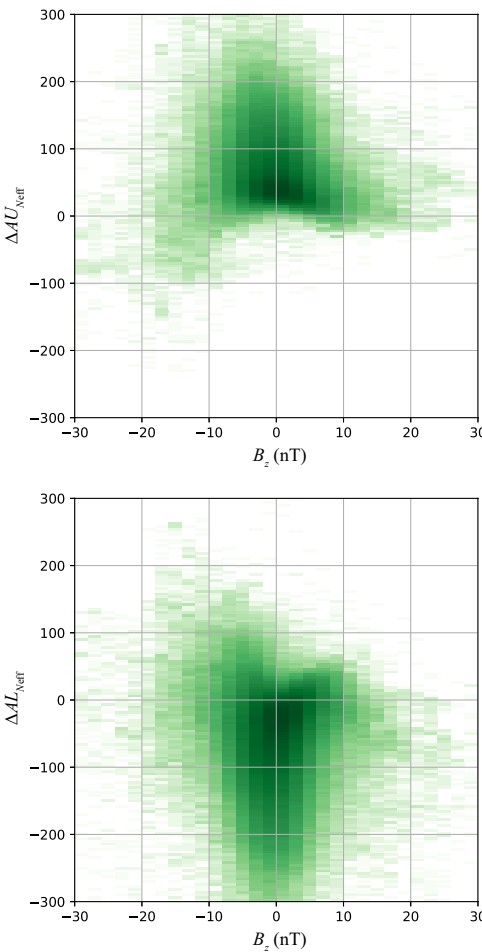

**Figure 11.** 2-dimensional histogram indicating the dependence of the solar-wind density effect on IMF $B_z$.

Although the solar-wind density effect is sometimes ignored when modeling the $AU$ and $AL$ indices, Gleisner and Lundstedy (1997) reported that the performance of a neural network for modeling the $AE$ index is improved by considering the solar-wind density effect. McPherron et al. (2015) also suggested a contribution from the solar wind density to the $AL$ index. Blunier et al. (2021) deduced the solar-wind parameters contributing to changes in the geomagnetic indices by using neural networks, and suggested that the solar-wind density has a more visible effect on $AU$ than on $AL$. The stronger effect on $AU$ suggested by Blunier et al. agrees with our result shown in Figure 6. Ebihara et al. (2019) conducted simulation experiments to examine the impact of various solar-wind parameters on the *SML* index (Newell and Gjerloev, 2011), which is an extension of the $AL$ index calculated with data from a larger number of observatories. According to their result, the *SML* index depends on the solar-wind density when IMF $B_z$ is weak, while it is not clearly affected by the solar-wind density when IMF $B_z$ is directed strongly southward. This simulation result is consistent with our result in Figure 11. Figure 11 may thus be regarded as statistical evidence of the compound effect between IMF $B_z$ and the solar-wind density.

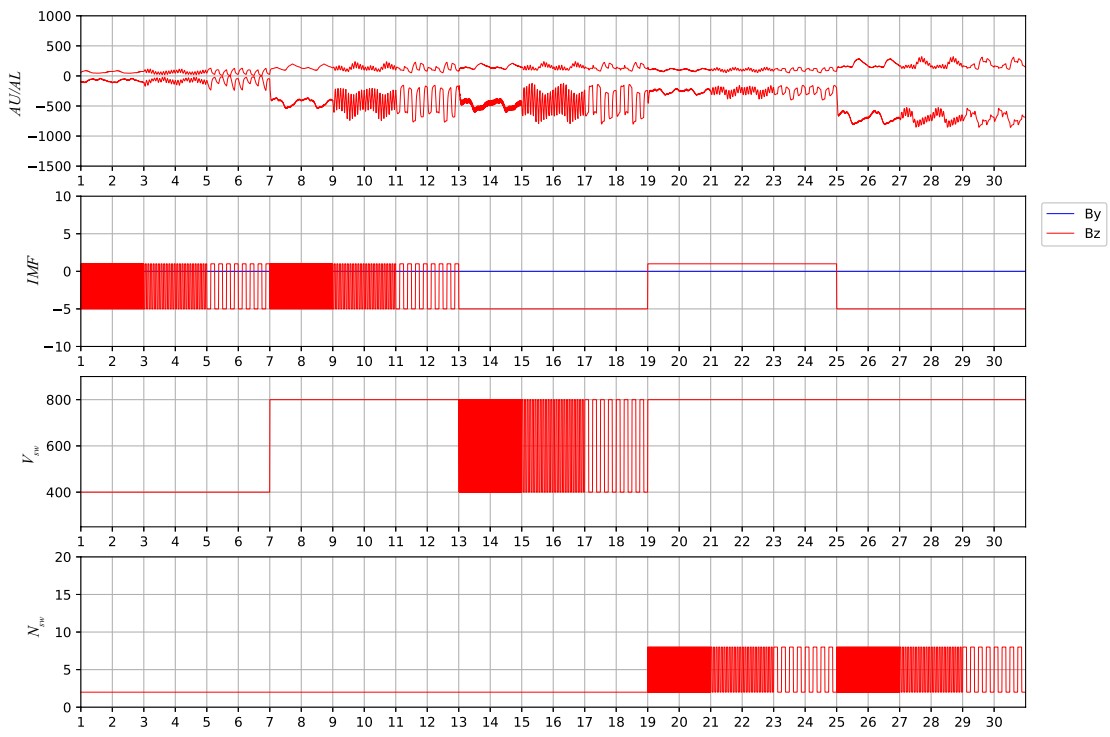

**Figure 12.** Result of an experiment in which the solar wind parameters are fixed at constant values except that one of the parameters are given by rectangular waves with various periods.

Figure 10 shows the compound effect between the solar wind density and velocity. One plausible explanation is the effect of the solar wind dynamic pressure which is proportional to $N_{sw}V_{sw}^2$. As some studies have suggested that field-aligned currents around the auroral latitudes are influenced by the solar-wind dynamic pressure (Iijima and Potemra, 1982; Wang et al., 2006; Nakano et al., 2009; Korth et al., 2010), it is possible that the enhancement of the field-aligned currents increases the auroral electrojets. Some studies suggested that the solar wind dynamic pressure makes a temporal effects on the ionospheric convection (Ober et al., 2007; Boudouridis et al., 2008). The convection enhancement could cause the increases of both $AU$ and $AL$. In particular, since the eastward electrojet represented by $AU$ is basically controlled by the ionospheric convection, the compound effect on $AU$ may be interpreted as the dynamic pressure effect. In Figure 10, however, the density effect on $AL$ disappears when the solar wind velocity is around $300\,\mathrm{km/s}$, while that on $AU$ is visible even under low solar-wind speed conditions. This can not be necessarily explained by the solar-wind dynamic pressure effect. This problem might be solved by considering the contribution of the plasma sheet condition. Sergeev et al. (2014, 2015) suggests that the plasma sheet temperature and density may affect the ionospheric conductivity in the region of the westward electrojet which the $AL$ index represents. It has been suggested that the plasma sheet temperature and density depend on the solar-wind velocity and density,

respectively (Terasawa et al., 1997; Nagata et al., 2007). The plasma sheet effect can thus partially contribute to the relationship between $AL$ and the solar-wind density.

## 6  Summary

This study modeled the temporal pattern of the $AU$ and $AL$ indices using ESN. Although the ESN model is relatively simple, it mostly accurately reproduces the variations of the $AU$ and $AL$ indices. We analyze the properties of the magnetospheric system by putting artificial inputs into the trained ESN model. Our results show a strong impact of the solar-wind speed which was previously observed in the literature. It is also suggested that IMF $B_y$ and the solar-wind density have significant effects, especially on the $AU$ index. These results are consistent with other studies. In addition, an analysis of the synthetic $AU$ and $AL$ indices obtained from the artificial inputs suggests that the solar-wind density does not have a simple linear effect on $AU$ and $AL$, but rather that some compound processes exist. According to the results, the solar-wind density contributes to the auroral electrojet intensity more effectively under high solar-wind speed conditions and the solar-wind density effect becomes small under low solar-wind speed conditions. The solar-wind density effect tends to be important when IMF $B_z$ is near zero. The density effect is small on average when $|B_z|$ is large.

*Data availability.*  The $AU$, $AL$, and *SYM-H* indices are available from the web site of the WDC for Geomagnetism, Kyoto (http://wdc.kugi.kyoto-u.ac.jp/wdc/Sec3.html). The OMNI solar wind data are available from the OMNIWeb of NASA/GSFC (https://omniweb.gsfc.nasa.gov/).

*Author contributions.*  Both authors built the research plan. SN conceived and conducted the analysis. RK contributed to the scientific interpretation.

*Competing interests.*  The contact author declare that any authors have no competing interests.

*Financial support.*  The work of SN was supported by JSPS KAKENHI (Grant Number 17H01704).

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
