# Peer review of "Echo state network model for analyzing solar-wind effects on the AU and AL indices"

_Annales Geophysicae, 2021_

## Author Comment (AC1)

We conducted an experiment with synthetic solar wind inputs where one of the parameters was varied in a square-wave form. The figure below shows the result of the experiment. IMF $B_z$ was repeatedly switched between $-5\,$nT and $1\,$nT with a period of 20 minute, 2 hours, and 6 hours in Day 1–2, 3–4, and 5–6 (the first, second, and third two days), respectively, while the solar wind speed was fixed at 400 km/s. Similarly, IMF $B_z$ was switched between $-5\,$nT and $1\,$nT with a period of 20 minute, 2 hours, and 6 hours in Day 7–8, 9–10, and 11–12, while the solar wind speed was constant at 800 km/s. From Day 13, IMF $B_z$ was fixed at $-5\,$nT. In Day 13–14, 15–16, and 17–18, the solar wind speed was switched between 400 km/s and 800 km/s with a period of 20 minute, 2 hours, and 6 hours, respectively. In Day 19–20, 21–22, and 23–24, the solar wind density was switched between 2 /cc and 8 /cc with a period of 20 minute, 2 hours, and 6 hours, respectively, while IMF $B_z$ was $1\,$nT. Again, the solar wind density was switched between 2 /cc and 8 /cc with a period of 20 minute, 2 hours, and 6 hours in Day 25–26, 27–28, and 29–30 while IMF $B_z$ was fixed at $-5\,$nT.

---

## Author Response (AR1)

**Response to the comments by Referee #1**

We appreciate the referee for the constructive comments and suggestions. We have revised the manuscript after carefully considering the comments raised by the reviewer. We have also corrected errors and deficiencies in Section 2 in this revision. In the following, the comments by the referee are quoted in Italic, and our reply is provided for each comment in Roman.

*Lines 102-103: "We can then identify properties of the auroral electrojets by analyzing the synthetic indices obtained from various artificial inputs"*

*This can be very useful, but it is surprising how it was done in practice. The construction of the "synthetic solar wind" was done by fixing one of the input parameters to zero or other fixed level in order to determine the effects on the output. While that process produces some useful results, I expected something like use of a step function for velocity or IMF values. For example, using a steady value of one parameter such as Bz for a period of time, such as a few hours, then stepping up to another value, and repeating. The results should show how the AU/AL indices respond to that parameter.*

According to the suggestion, we conducted some experiments which give a variation of a solar wind parameter by as rectangular waves with various periods. The results of the experiments are shown in Figure 12. It is not guaranteed that the ESN output reflects on the actual response time scale because the ESN output tends to be smoother than the observed variation as shown in Figures 8 and 9 (in the revised version). However, many of the characteristics of the result seem to be consistent with the results observed in Figure 8-11 (in the revised version).

*Line 94 and elsewhere: "ESN meets a satisfactorily high accuracy". I think the accuracy is overstated, as the model output seems to miss the amplitudes of a lot of AU/AL variations.*

We agree with the referee that the model output tends to underestimate the amplitudes of impulsive variations. We intended to just point out that the ESN achieve a comparable accuracy to existing models. We have modified the expression (L. 109).

*Table 1 and elsewhere in the text: It would be helpful to include correlations, as another measure of model performance.*

The correlation coefficients have been added in the revised version of Table 1. In addition, we found that the *AU* and *AL* data in 1998 and 1999 were not correctly processed in the previous version, and the RMSE values in 1998 and 1999 were also revised.

*The paper needs to have more details in the Discussion or Summary regarding the relationship of the results with the dynamic pressure (nV2/2). There is no mention of pressure, although it appears in several previous publications, such as Newell et al. [2007 and 2008], as related to the effects of the density. Interestingly, the results in Figure 9 seem to follow a V2 curve.*

*I think that two of the references cited by Newell et al. [2008] had indicated that sudden increases in dynamic pressure only produced on temporary response, in the magnetosphere [Boudouridis et al, 2005, Ober et al., 2007]. For example, the polar cap electric potential may increase for a while, then go back down to near the pre-impulse level. This temporary behavior complicates any search for a consistent relationship between the solar wind density and ionospheric response.*

We already mentioned that the dynamic pressure effect possibly explains Figure 9 (Figure 10 in the revised version) (Lines 182-186). As we discussed in Lines 217–219, however, our opinion is that the dynamic pressure effect does not completely explain the compound effect between the solar wind density and velocity. As shown in the lower panel of Figure 10 in the revised version, the density effect on AL becomes zero on average when the solar wind velocity is around 300 km/s. If the density effect was totally due to the dynamic pressure effect, the density effect would be visible even under the low-speed solar wind conditions.

However, the upper panel of Figure 10 suggests that the density effect on AU is non-zero even if the solar wind speed is small. This result may be explained as the dynamic pressure effect. It would be possible that AU is related with the polar cap potential which is affected by the solar wind dynamic pressure as suggested by the referee. We have added the discussion on the possible relationship between AU and the dynamic pressure in the revised version (L. 214–217).

*In Figure 1, the graph showing the three IMF components is not clear. These should be put into three separate rows.*

Our opinion is that the differences among the three IMF components are not very important in this figure. We just intend to show the correspondence between IMF fluctuations and auroral activities in this figure. It would be enough if the three IMF components can be distinguished in Figure 8 and 9 (in the revised version).

*Lines 98-99: This sentence is not clear.*

This sentence has been removed to avoid the confusion.

*Figure 3 needs to be taller in order to help show the differences between some of the lines.*

We appreciate for the comment. We have added Figure 5 to clearly show the effects of $B_y$ and $N_{sw}$ in the revised version.

> *I don't agree with the use of the word "sounding," and a different terminology would be preferred as the title sounds a little pretentious. In my opinion, this use doesn't agree with any of the multiple, dictionary meanings of the word "sound" or "sounding." Whether or not a change is made is entirely up to the discretion of the authors.*

We have modified the title. The word "sound" is not used in the revised version.

> *The web link for WDC for Geomagnetism, Kyoto isn't working, due to the line break. A different Latex package for URL references might work, or try putting the link all on one line without a break. This is a common problem encountered with URLs in Latex.*

We appreciate the referee for pointing out the problem. We think that this problem can be fixed when typesetting the final version.

**Response to the comments by Referee #2**

We appreciate the referee for the constructive comments and suggestions. We have revised the manuscript after carefully considering the comments raised by the reviewer. We have also corrected errors and deficiencies in Section 2 in this revision. In the following, the comments by the referee are quoted in Italic, and our reply is provided for each comment in Roman.

*Introduction: To increase the interest of his paper for general audience the authors should give a bit broader explanation what are auroral indices (what current systems they try to measure) and how previous studies have found that solar wind properties control them (parameters that are the most important and why). Discussion has partly this information, but could be already here.*

We have added some explanations on the auroral indices (L. 10–12, L. 15–20). We appreciate the referee for the constructive suggestion.

*Introduction: Authors could also discuss in the Introduction why they expect to detect non-linearities.*

Most of widely-used machine learning models are designed for representing nonlinearities. An echo state network (ESN) can also be used for predictions of nonlinear systems because it contains nonlinear functions as described in Eq. (1). Indeed, Chattopadhyay et al. (2020), which is cited in the introduction section, shows an example which represents nonlinear dynamics by an ESN. In the revised version, we have added a mention that an ESN is used for applying to various nonlinear problems (L. 37–38).

*Pages 72: Is this now meant to take into account the timelag between solar wind parameters and AL/AU response? What is the typical timelag giving the best result? Also the optimal timelag could vary depending on the solar wind parameter in question, could that have an effect to the results or their interpretation.*

Here we intend to say that the ESN refers to a time sequence of the input data for making a prediction. Each of the state variables of the ESN is obtained by a nonlinear conversion of the previous state variables according to Eq. (1). The ESN thus keeps the history of the input data in memory. A prediction by the ESN is based on the history of the data. In order to predict something using the ESN, a sufficient time sequence of the data must be fed as an input in advance. We have added an explanation on how the ESN works (L. 83–85).

The timelag between the input and output would be learned when training the ESN model. However, the timelag is not given by a specific parameter in the ESN. It is difficult to quantify the timelag using the trained ESN model.

*Figure 1 discussion: It seems that the model consistently underestimates the observed values both for AU and AL. What is the expected reason to this.*

There is no expected reason to underestimate the observed values. Our ESN model is tuned so that the mean difference between the model output and observation is zero for the period from 2005 to 2014. Figure 4 and 7 show that the mean difference is very small ($\sim 10\,\mathrm{nT}$ when using all the solar-wind parameters) even when it is applied for the period from 1998 to 2004. However, since the ESN model cannot represent short-term variations in detail, the ESN output tends to be much smoother than the observation. The amplitudes of the short-term variations thus tend to be reduced in the model output. Maybe this is the reason why it looks as if the model underestimates the observation.

*Figure 1: Top panel label is AE, should rather be AL & AU?*

We agree with the referee. We have revised the label.

*Page 138: Does synthetic mean here that the AL and AU show are produced using solar wind observed gathered 21 October to 25 October in 1999? Would it be better call it modelled than synthetic?*

In Figure 8 (Figure 7 in the previous version), the red line shows the model output obtained with the observed solar wind data. On the other hand, the green and blue lines were obtained with artificial inputs where one of the solar-wind parameters was fixed at a constant. The main purpose of this figure is to show the results with synthetic inputs, and we refer to them as the synthetic AU and AL indices.

*Figure 7: colors are not well visible here in the top panel. E.g., I cannot see any green line.*

We have changed the color of the green lines in Figures 7 and 8 (Figures 8 and 9 in the revised version). We appreciate for the comment.